# Effectiveness of Exercise, Cognitive Behavioral Therapy, and Pharmacotherapy on Improving Sleep in Adults with Chronic Insomnia: A Systematic Review and Network Meta-Analysis of Randomized Controlled Trials

**DOI:** 10.3390/healthcare11152207

**Published:** 2023-08-04

**Authors:** Danny J. Yu, Francesco Recchia, Joshua D. K. Bernal, Angus P. Yu, Daniel Y. Fong, Shirley X. Li, Rachel N. Y. Chan, Xiaoqing Hu, Parco M. Siu

**Affiliations:** 1Division of Kinesiology, School of Public Health, Li Ka Shing Faculty of Medicine, The University of Hong Kong, Hong Kong, China; juchengyu@cuhk.edu.hk (D.J.Y.); u3007503@connect.hku.hk (F.R.); jdkb9701@connect.hku.hk (J.D.K.B.); phayu@hku.hk (A.P.Y.); 2Li Chiu Kong Family Sleep Assessment Unit, Department of Psychiatry, Faculty of Medicine, The Chinese University of Hong Kong, Hong Kong, China; rachel.chan@cuhk.edu.hk; 3School of Nursing, Li Ka Shing Faculty of Medicine, The University of Hong Kong, Hong Kong, China; dytfong@hku.hk; 4Department of Psychology, Faculty of Social Sciences, The University of Hong Kong, Hong Kong, China; shirleyx@hku.hk (S.X.L.); xiaoqinghu@hku.hk (X.H.); 5The State Key Laboratory of Brain and Cognitive Sciences, The University of Hong Kong, Hong Kong, China

**Keywords:** chronic insomnia, exercise, cognitive-behavioral therapy for insomnia (CBT-I), pharmacotherapy, network meta-analysis

## Abstract

Despite the well-established treatment effectiveness of exercise, cognitive behavioral therapy for insomnia (CBT-I), and pharmacotherapy on improving sleep, there have been no studies to compare their long-term effectiveness, which is of clinical importance for sustainable management of chronic insomnia. This study compared the long-term effectiveness of these three interventions on improving sleep in adults with chronic insomnia. MEDLINE, PsycINFO, Embase, and SPORTDiscus were searched for eligible reports. Trials that investigated the long-term effectiveness of these three interventions on improving sleep were included. The post-intervention follow-up of the trial had to be ≥6 months to be eligible. The primary outcome was the long-term effectiveness of the three interventions on improving sleep. Treatment effectiveness was the secondary outcome. A random-effects network meta-analysis was carried out using a frequentist approach. Thirteen trials were included in the study. After an average post-intervention follow-up period of 10.3 months, both exercise (SMD, −0.29; 95% CI, −0.57 to −0.01) and CBT-I (−0.48; −0.68 to −0.28) showed superior long-term effectiveness on improving sleep compared with control. Temazepam was the only included pharmacotherapy, which demonstrated superior treatment effectiveness (−0.80; −1.25 to −0.36) but not long-term effectiveness (0.19; −0.32 to 0.69) compared with control. The findings support the use of both exercise and CBT-I for long-term management of chronic insomnia, while temazepam may be used for short-term treatment.

## 1. Introduction

Chronic insomnia is the most common sleep disorder in the general population [1], with a worldwide prevalence rate ranging from 3.9% to 22.1% depending on the diagnostic criteria used [2,3,4,5]. Insomnia increases the risk of developing various physical and mental disorders, including cardiovascular diseases, depression, and cognitive impairment [6]. Insomnia is also associated with increased health-care utilization and high economic and societal burdens [7,8]. Currently, cognitive-behavioral therapy for insomnia (CBT-I) and pharmacotherapy are recognized as common treatment approaches for managing chronic insomnia [9,10]. Both CBT-I and pharmacotherapy have been demonstrated to improve various sleep parameters, such as sleep latency, number of awakenings, wake time after sleep onset, total sleep time, and sleep quality [11]. However, CBT-I is not readily available due to the need for well-trained therapists and the high treatment costs, which limits its accessibility and scalability in the community [11,12]. Pharmacotherapies also have well-known limitations, including side effects, concerns of drug dependence and tolerance, and low acceptance levels [13,14,15]. There is a need to identify alternative treatments for insomnia that are readily accessible with promising therapeutic effectiveness and low adverse effects. Exercise has been proposed as an alternative non-pharmacological approach for treating chronic insomnia [16]. The World Health Organization (WHO) guidelines recommend that adults regularly exercise to improve sleep [17]. Similarly, the American Physical Activity Guidelines recommend that adults engage in regular exercise to improve sleep outcomes, including sleep efficiency, sleep quality, daytime sleepiness, and use of sleep medications [18]. Recent meta-analyses have shown that different exercise modalities can improve sleep, such as aerobic exercise, resistance training and mind–body exercise [16,19,20,21]. A previous randomized controlled trial reported that a 4-month Tai Chi intervention significantly improved subjective sleep quality in patients with chronic insomnia [22]. Another randomized controlled trial demonstrated that a 12-week program of aerobic exercise and muscle-strengthening training effectively reduced the severity of insomnia [23]. However, it is less clear whether the treatment effectiveness of exercise on improving sleep is comparable to that of CBT-I or pharmacotherapy [11,24]. Validating the treatment effectiveness of exercise is of clinical practice importance for recognizing it as an alternative non-pharmacological approach to treat chronic insomnia. Moreover, there has been less emphasis on developing interventions for sustainable management of chronic insomnia [25]. Longitudinal studies have reported that the median duration of chronic insomnia is 3 years, with 56% to 74% of chronic insomnia patients having experienced persistent symptoms within the past year [2,26,27,28]. More alarmingly, 27% of patients experience a relapse post-remission [28]. Ideally, treatments should provide long-term effectiveness to help patients manage their chronic insomnia symptoms [29]. Although several studies have evaluated the treatment effectiveness of exercise, CBT-I, and pharmacotherapy on improving sleep [11,16,24], no studies have compared the long-term effectiveness of these three approaches in patients with chronic insomnia, which is of practical importance for identifying sustainable solutions to chronic insomnia management. Using a network meta-analytic approach, we compared the treatment effectiveness and long-term effectiveness (≥6 months post-intervention follow-up [30]) of exercise, CBT-I, and pharmacotherapy on improving sleep in adults with chronic insomnia.

## 2. Materials and Methods

This study adhered to the Preferred Reporting Items for Systematic Reviews and Meta-Analyses (PRISMA) extension guidelines for network meta-analyses [31] and the review protocol was registered in PROSPERO (CRD42022296586).

### 2.1. Data Sources and Searches

We systematically searched MEDLINE, PsycINFO, Embase, and SPORTDiscus for relevant articles published from the database inception to 30 January 2022. A detailed description of the search strategy is provided in Appendix A. Two independent researchers (Danny J. Yu and Francesco Recchia) performed the search using pre-established criteria. A third researcher (Joshua D.K. Bernal) was consulted in case of disagreements, which were resolved by consensus among all three researchers.

### 2.2. Study Selection

#### 2.2.1. Type of Studies

We included randomized controlled trials investigating both the treatment effectiveness and long-term effectiveness of exercise, CBT-I, and pharmacotherapy on improving sleep in adults with chronic insomnia. As the intervention duration might vary in different settings and we wanted to examine the treatment effectiveness of the three approaches from a wide range of clinical practices, there were no restrictions on the intervention duration of the study, which is the time duration between the baseline measurement and the post-intervention measurement. To assess the long-term effectiveness, the post-intervention follow-up period of the study, which is the time duration between the post-intervention measurement and the last follow-up measurement, had to be ≥ 6 months to be eligible.

#### 2.2.2. Types of Participants

The study participants were adults with chronic insomnia aged ≥18 years. Chronic insomnia was defined as having difficulty in initiating sleep, maintaining sleep, or with early morning awakening, with complaints of impaired daytime functioning and the sleep difficulty occurring at least three nights per week and lasting for at least three months, which is in accordance with standard diagnostic criteria, such as the Diagnostic and Statistical Manual of Mental Disorders, Fifth Edition, and the International Classification of Sleep Disorders, Third Edition [32,33,34]. Studies involving insomnia patients with common comorbidities such as depression, anxiety, and fibromyalgia were also included to enhance the generalizability of the findings. However, studies that involved participants with comorbid conditions that would prevent them from joining the exercise intervention, such as severe musculoskeletal disorders or dementia, were excluded to avoid violation of the transitivity assumption [35].

#### 2.2.3. Types of Interventions

Exercise was defined as “planned, structured, and repetitive bodily movement aimed to improve and/or maintain one or more components of physical fitness” according to the American College of Sports Medicine guidelines [36]. CBT-I was defined as a multimodal approach incorporating at least two of following components: cognitive therapy, stimulus control, sleep restriction, sleep hygiene, and relaxation therapy [10,24]. To stringently compare the treatment effectiveness and long-term effectiveness of exercise to that of CBT-I, we excluded studies with online or self-help CBT-I interventions, which are commonly recognized as effective but not fully comparable as traditional face-to-face CBT-I interventions [37,38]. Studies with online or unsupervised exercise interventions were subsequently excluded to avoid violation of the transitivity assumption [35]. Lastly, pharmacotherapy was defined as a pharmacological intervention using any of the eight sleep-promoting agents (suvorexant, eszopiclone, zaleplon, zolpidem, triazolam, temazepam, ramelteon and doxepin) recommended by the American Academy of Sleep Medicine (AASM) clinical practice guidelines [9]. 

### 2.3. Outcome Measures

The primary outcome of this study was the long-term effectiveness of the three interventions on improving sleep, which reflected the sleep-promoting effects of the interventions during the post-intervention follow-up period and was operationalized as the score of the validated scale used to subjectively measure sleep improvement at the last post-intervention follow-up measurement. The secondary outcome was the treatment effectiveness of the interventions on improving sleep, which reflected the sleep-promoting effects of the interventions during the intervention period and was operationalized as the score of the validated scale used to subjectively measure sleep improvement at the post-intervention measurement. When multiple validated scales were used in the study, a reductive meta-analytic approach was applied to randomly select one of the scales for data synthesis and to yield a representative effect for the network meta-analysis [39,40].

### 2.4. Data Extraction and Quality Assessment

Two independent researchers extracted the samples sizes, scores, and standard deviations of the subjective sleep assessment scales in each trial. Any disagreements on the extracted data were resolved by consensus. We also extracted information on participant characteristics (e.g., age, sex, and comorbidity) and trial characteristics (e.g., first author, type of intervention and control, intervention and follow-up duration) using Covidence, a web-based systematic review tool [41]. 

### 2.5. Risk of Bias and Confidence Assessment

Risk of bias was rated using the Cochrane risk of bias assessment tool (RoB-2), which comprehensively assessed five components: randomization process, deviations from intended interventions, missing outcome data, measurement of the outcome (measurement appropriateness and blindness), and selection of the reported results [42]. Confidence of the accumulated evidence in the network was assessed by the Confidence in Network Meta-Analysis (CINeMA), a web application of the Grading of Recommendations, Assessment, Development, and Evaluations (GRADE) ratings approach [43,44]. Additional information on RoB-2 and CINeMA are provided in Appendix A.

### 2.6. Data Synthesis and Analysis

We conducted a network meta-analysis with a frequentist framework using the netmeta package in the statistical software R (version 4.1.2). The network included: (1) exercise, (2) CBT-I, (3) pharmacotherapy, and (4) control. We used a random-effects pairwise meta-analysis to estimate standardized mean differences (SMD) for direct comparisons. Indirect evidence was assessed using the network. The random effects netmeta model was used to control for multi-arm randomized controlled trials. Results of the primary outcome (long-term effectiveness) and secondary outcome (treatment effectiveness) were expressed as SMDs with 95% confidence intervals. The pairwise between-study heterogeneity was examined using Cochran’s Q statistics. In addition, T^2^ was calculated to determine the level of variance between studies, and I^2^ was used to determine the percentage of variance due to between-study heterogeneity. 

### 2.7. Transitivity Assessment

The transitivity assumption was tested by assessing the distribution of patients and intervention characteristics across the comparisons [35]. The statistical manifestation of transitivity–consistency among the comparisons was also measured to assess the agreement of direct and indirect evidence in the network [35]. Local and global approaches were applied to measure consistency using netsplit and decomp.design functions, respectively. We presumed that every participant in the included studies could potentially be randomized to any of the treatments compared.

### 2.8. Sensitivity Analysis

Three sensitivity analyses were performed to test the robustness of the results. First, we extended the minimal follow-up length from 6 months to 12 months by including studies with a follow-up period ≥12 months to examine the longer-term effectiveness of the three interventions. Second, we excluded studies with high level of indirectness. Last, we included all the data generated by every scale in the included studies. Given the small number of studies in each comparison group, subgroup and meta-regression analyses could not be performed to further explore potential sources of heterogeneity.

### 2.9. Role of the Funding Source

This study was supported by General Research Fund of Research Grants Council (RGC), Hong Kong University Grants Committee (project number: 17112819) and Seed Fund for Basic Research of the University of Hong Kong. The funding bodies had no role in the study design, data collection, analysis and interpretation, report writing, or the decision to submit for publication.

## 3. Results

### 3.1. Overview of Studies

The literature search identified 9447 potential studies. After excluding duplicate studies, 6674 studies were screened by title and abstract, and 6449 studies were excluded. In total, 225 full-text articles were retrieved for in-depth screening. In total, 212 studies were excluded due to a post-intervention follow-up period <6 months (*n* = 136), ineligible study design such as crossover study (*n* = 42), secondary analysis (*n* = 21), ineligible population (*n* = 9) and ineligible outcome (*n* = 4). A total of 13 studies were included in the main analysis (Appendix A). The 13 studies included 1350 participants in 18 pairwise comparisons among the three treatment and control groups (Figure 1). The 18 pairwise comparisons included: exercise vs. control (*n* = 3), CBT-I vs. control (*n* = 10), and pharmacotherapy vs. control (*n* = 1), exercise vs. CBT-I (*n* = 2), and CBT-I vs. pharmacotherapy (*n* = 2).

Exercise interventions included tai chi exercise [22,23,45] and aerobic and muscle-strengthening exercise [23]. All CBT-I interventions included at least cognitive therapy, stimulus control, and sleep restriction [22,45,46,47,48,49,50,51,52,53,54,55]. All pharmacotherapy interventions prescribed temazepam [53,55]. The control interventions included sleep education, sleep hygiene, relaxation training, usual care, and placebo. The intervention durations ranged between 5 and 16 weeks and the post-intervention follow-up durations ranged between 6 and 24 months. The outcome measurements included Athens insomnia scale (AIS) [56], insomnia severity index (ISI) [57], insomnia symptom questionnaire (ISQ) [58], Pittsburgh sleep quality index (PSQI) [59], and sleep diary (sleep efficiency) [60]. AIS, ISI, and ISQ measured the perceived insomnia severity. PSQI and the sleep efficiency domain of the sleep diary reflected the subjective sleep quality. These five instruments are commonly used in research and clinical settings to subjectively assess the sleep improvement after treatments.

### 3.2. Risk of Bias and Confidence

One study was assessed to have a low risk of bias and 12 studies had some concerns, whereas no study was assessed to have a high risk of bias (Appendix A). Most of the network evidence relied on studies with a low to moderate level of risk of bias and indirectness (Appendix A). The CINeMA report showed a high level of confidence for all the comparisons in the network (Appendix A).

### 3.3. Transitivity

The plausibility of the transitivity assumption was examined by assessing the eligibility criteria, the patient characteristics, and the study designs. The major patient characteristics and intervention characteristics did not significantly differ among studies (Table 1, and Appendix A). Overall, the assumption of transitivity was valid, and we presumed that every participant in the included studies could potentially be allocated to any of the compared interventions.

### 3.4. Primary Outcome—Long-Term Effectiveness

After an average post-intervention follow-up duration of 10.3 months (95% CI, 6.7 to 13.9), exercise (SMD, −0.29; 95% CI, −0.57 to −0.01, I^2^ = 55.6%) and CBT-I (SMD, −0.48; 95% CI, −0.68 to −0.28, I^2^ = 55.6%) were superior on improving sleep compared to the control, whereas pharmacotherapy (temazepam) did not demonstrate any sustained beneficial effects (SMD 0.19; 95% CI, −0.32 to 0.69, I^2^ = 55.6%) (Table 2).

The pairwise analysis supported the findings of the network analysis, with both exercise (SMD, −0.39; 95% CI −0.71 to −0.07, I^2^ = 0.0%) and CBT-I (SMD, −0.43; 95% CI −0.74 to −0.12, I^2^ = 68.4%) eliciting greater improvements on sleep compared to the control, whereas pharmacotherapy (temazepam) showed no beneficial effects (SMD, −0.17; 95% CI −0.84 to 0.50, I^2^ not available) (Table 2). There was a moderate level of heterogeneity (T^2^ = 0.06; I^2^ = 55.6%) detected in the network [39], which could be mostly explained by heterogeneity among the different study designs (Q = 30.16; d.f. = 13; *p* = 0.005). The pairwise comparisons of the study designs included exercise vs. control, CBT-I vs. control, pharmacotherapy vs. control, exercise vs. CBT-I, and CBT-I vs. pharmacotherapy. After decomposition of the study designs, the comparison of CBT-I vs. control was found to have the most heterogeneity (Q = 28.40; d.f. = 9; *p* = 0.001). There was no evidence of inconsistency among the study designs (Q = 3.62; d.f. = 2; *p* = 0.163). 

### 3.5. Secondary Outcome—Treatment Effectiveness

After an average 8.5-week intervention (95% CI, 6.7 to 10.2), exercise (SMD, −0.44; 95% CI −0.65 to −0.22, I^2^ = 32.4%), CBT-I (SMD, −0.60; 95% CI −0.76 to −0.44, I^2^ = 32.4%), and pharmacotherapy (temazepam) (SMD, −0.80; 95% CI −1.25 to −0.36, I^2^ = 32.4%) elicited superior effects on improving sleep compared to the control (Appendix A).

### 3.6. Sensitivity Analysis

Sensitivity analyses confirmed the robustness of the results. Both exercise and CBT-I showed sustained sleep-promoting effects in all three sensitivity analyses (Appendix A).

## 4. Discussion

This is the first network meta-analysis to comparatively evaluate the treatment effectiveness and long-term effectiveness of exercise, CBT-I, and pharmacotherapy on improving sleep in adults with chronic insomnia. Our results demonstrated that both exercise and CBT-I showed superior long-term effectiveness on improving sleep compared to the control, while pharmacotherapy (temazepam) showed excellent treatment effectiveness. Exercise demonstrated significant sleep-promoting effects, with a small-to-medium effect size compared to the control (SMD, −0.29) after an average 10.3-month post-intervention follow-up [61]. Takemura et al. reported that the sleep-promoting effects of aerobic exercise remained significant (SMD, −0.37; 95% CI, −0.55 to −0.18) for 3 to 6 months post-intervention in cancer patients with poor sleep quality [62], whereas our meta-analysis demonstrated that the sleep-promoting effects of exercise can last much longer. In the CINeMA confidence assessment, the selection of a clinically significant effect size of −0.4 was based on a previous meta-analysis [30], in which CBT-I was shown to elicit long-term improvements on sleep after ≥6-month post-intervention follow-up when compared to the control [30]. In the current meta-analysis, although the long-term effectiveness of exercise did not reach the pre-defined clinically significant effect size of −0.4, exercise was found to elicit a greater hypnotic effect (SMD, −0.32) when the minimal follow-up period was extended from 6 months to 12 months (average follow-up length increased from 10.3 to 16.8 months). More importantly, there were no significant differences between the long-term effectiveness of exercise and that of CBT-I. Considering the limited accessibility and scalability of CBT-I in the community, our study supports the use of exercise as an alternative non-pharmacological treatment for the long-term management of chronic insomnia in adults [25]. At the end of the interventions, all three treatments were effective in improving sleep compared to the control. Pharmacotherapy (temazepam) showed the greatest magnitude of improvement (SMD, −0.80), followed by CBT-I (SMD, −0.60) and exercise (SMD, −0.44). We appraised the magnitudes of improvement with previous literature to assess whether the results are representative of the existing exercise, CBT-I and pharmacotherapy trials. Milne et al. included 6 exercise interventional studies in a meta-analysis and found that exercise significantly improved sleep (SMD, −0.47; 95% CI, −0.86 to −0.08) [16]. One previous meta-analysis by Koffel et al. summarized the treatment effectiveness of CBT-I (SMD, −0.85; 95% CI, −0.57 to −1.14) from 8 trials [63]. Pharmacotherapy was also found to be able to significantly improve sleep after the intervention (SMD, −0.87; 95% CI, unreported) in a meta-analysis with 21 studies included [16]. Indeed, the magnitudes of improvements found in our study are in line with the previous meta-analyses [11,16,63], suggesting that although only 13 trials were included in our study, these studies are still representative of the existing literature. More importantly, using a network meta-analytical approach, this is the first study demonstrated that exercise had comparable treatment effectiveness to that of CBT-I (SMD, 0.16; 95% CI, −0.07 to 0.40). Our findings support current physical activity guidelines recommending regular exercise in adults to improve sleep [17,18].

### Limitations

This meta-analysis had several limitations that should be noted. First, only two pharmacotherapy studies were included and both studies prescribed temazepam as the intervention [53,55], which limited the generalizability of our findings. The current results showing that temazepam did not demonstrated significant long-term effectiveness on improving sleep must be interpreted with caution and cannot be generalized to other pharmacotherapies, such as suvorexant, eszopiclone, zaleplon, zolpidem, triazolam, ramelteon and doxepin. During the full-text screening of eligible studies, we found that a considerable number of high-quality studies reported the superior treatment effectiveness of different pharmacotherapy interventions over placebo [64,65,66,67,68,69,70], and several meta-analytical studies have also proved the therapeutic effects of sleep medications [11,71,72]. However, it is regrettable to point it out that most of the pharmacotherapy intervention studies did not set a post-intervention follow-up, or only designed with a relatively short post-intervention follow-up ranging from 24 h to 3 months, which largely limited the current investigation of the long-term effectiveness of pharmacotherapy on improving sleep. In line with previous review articles by Morin et al. [25] and Perlis et al. [73], our study further added evidence showing that the sleep-promoting effects of pharmacotherapy, specifically temazepam, were not sustainable after ≥ 6 months post-intervention follow-up, and highlighting the urgent need for future studies to delineate the long-term effectiveness of pharmacotherapy interventions.

Another limitation of this study is that besides one study investigated the sleep-promoting effects of aerobic exercise and resistance training [23], the exercise interventions in the other two included studies were both Tai Chi-based [22,45], which might prevent our current conclusion to be extended to different exercise modalities. Nevertheless, one recent network meta-analysis summarized the comparative treatment effectiveness of different exercise modalities on improving sleep quality in older adults, and the results showed that Tai Chi had similar sleep-promoting effects as 7 other different exercise regimens, including yoga, Pilates, walking, muscle endurance training, muscle endurance training combined with walking, resistance training and resistance training combined with walking [21]. We believe that the long-term sleep-promoting effects of Tai Chi will also be comparable to those of other exercise modalities. One of our previous works demonstrated that the aerobic and muscle-strengthening exercise exerted similar long-term beneficial effects to Tai Chi on improving sleep [23]. Future studies are warranted to investigate the comparative long-term effectiveness of different exercise modalities.

The third limitation is that only subjective sleep measurements were included in the study, which might inherit the risk of self-report bias. Nevertheless, it should be noted that the diagnosis of chronic insomnia heavily relies on subjective sleep assessments [32]. The AASM clinical guidelines for the evaluation and management of chronic insomnia in adults suggests that subjective sleep assessments aid baseline evaluation and outcome follow-up in patients with chronic insomnia [1]. Evaluating the long-term effectiveness and treatment effectiveness of exercise, CBT-I, and pharmacotherapy on improving sleep using subjective sleep measurements is valid and in accordance with clinical practice.

## 5. Conclusions

This network meta-analysis showed that both exercise and CBT-I were effective in improving sleep in adults with chronic insomnia in the long term (≥ 6-month post-intervention follow up), while the long-term effectiveness of temazepam was not superior to that of the control. All three interventions demonstrated excellent treatment effectiveness. These findings support the use of both exercise and CBT-I for long-term management of adults with chronic insomnia, and that temazepam may be used for short-term insomnia treatment.

## Figures and Tables

**Figure 1 healthcare-11-02207-f001:**
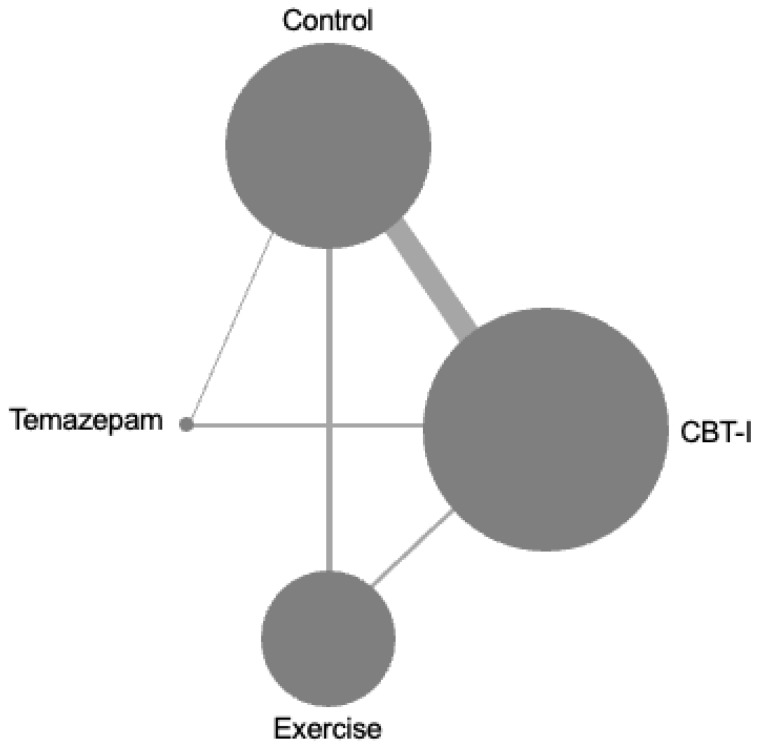
Geometry of the network. The size of the node represents the number of participants in the intervention. The thickness of the edges represents the number of studies in each treatment comparison.

**Table 1 healthcare-11-02207-t001:** Characteristics of the included studies.

Study	Mean Age (SD)	Gender (Female %)	Clinical Condition	Assessment Instrument	Intervention	Follow-Up Duration ^#^
Alessi, 2016 ^b^	72 (8)	3.1%	Chronic insomnia	ISI PSQI * Sleep diary	CBT-I Sleep education	12
Clarke, 2015 ^b^	52 (13)	68.8%	Depressive patients with chronic insomnia	ISI	CBT-I Relaxation training	6
Edinger, 2005 ^b^	48 (8)	95.7%	Fibromyalgia patients with chronic insomnia	ISQ Sleep diary *	CBT-I Sleep hygiene	6
Espie, 2007 ^b^	54 (15)	68.2%	Chronic insomnia	PSQI * Sleep diary	CBT-I Usual care	6
Espie, 2008 ^b^	61 (12)	68.7%	Cancer survivors with chronic insomnia	Sleep diary	CBT-I Usual care	6
Edinger, 2009 ^b^	54 (14)	26.8%	Chronic insomnia	ISQ PSQI *	CBT-I Sleep hygiene	6
Irwin, 2014 ^a,b,d^	66 (7)	71.5%	Chronic insomnia	AIS * PSQI Sleep diary	CBT-I Tai Chi Sleep education	12
Irwin, 2017 ^d^	60 (9)	100%	Breast cancer survivors with chronic insomnia	AIS PSQI * Sleep diary	CBT-I Tai Chi	12
Järnefelt, 2020 ^b^	42 (30)	74.6%	Chronic insomnia	ISI Sleep diary *	CBT-I Sleep hygiene	6
Morin, 1999 ^e^	64 (7)	62.9%	Chronic insomnia	Sleep diary	CBT-I Temazepam	24
Martínez, 2014 ^b^	48 (7)	100%	Fibromyalgia patients with chronic insomnia	PSQI	CBT-I Sleep hygiene	6
Siu, 2021 ^a^	67 (7)	80.0%	Chronic insomnia	ISI * PSQI Sleep diary	Tai Chi Conventional exercise Usual care	24
Wu, 2006 ^b,c,e^	38 (12)	53.2%	Chronic insomnia	Sleep diary	CBT-I Temazepam Placebo	8

^a^: included in the Exercise vs. Control comparison; ^b^: included in the CBT-I vs. Control comparison; ^c^: included in the Pharmacotherapy vs. Control comparison; ^d^: included in the Exercise vs. CBT-I comparison; ^e^: included in the CBT-I vs. Pharmacotherapy comparison. ISI: Insomnia Severity Index, PSQI: Pittsburgh Sleep Quality Index, ISQ: Insomnia Symptom Questionnaire; AIS: Athens Insomnia Scale. * The results of the scale were randomly selected for data synthesis and analysis. ^#^: The unit of the follow-up duration is expressed in months.

**Table 2 healthcare-11-02207-t002:** Results on the comparative long-term effectiveness of the interventions from the network and pairwise meta-analyses.

**Exercise**	0.36 (−1.73 to 2.46) (N = 2; I^2^ = 20.1%)	NA^2^	**−0.39** (−0.70 to −0.07) (N = 2; I^2^ = 0.0%)
0.19 (−0.11 to 0.49) (N = 13; I^2^ = 55.6%)	**CBT-I**	**−0.85** (−1.23 to −0.48) (N = 2, I2 = 0.0%)	**−0.43** (−0.74 to −0.12) (N = 10, I^2^ = 68.4%)
−0.48 (−1.03 to 0.08) (N = 13; I^2^ = 55.6%)	**−0.66** (−1.15 to −0.18) (N = 13; I^2^ = 55.6%)	**Temazepam**	−0.17 (−0.84 to 0.50) (N = 1; I^2^ = NA^1^)
**−0.29** (−0.57 to −0.01) (N = 13; I^2^ = 55.6%)	**−0.48** (−0.68 to −0.28) (N = 13; I^2^ = 55.6%)	**0.19** (−0.32 to 0.69) (N = 13; I^2^ = 55.6%)	**Control**

Results of the network meta-analyses are presented in grey boxes, and results of the pairwise meta-analyses are presented in white boxes. Estimates are displayed as column vs. row for the network meta-analyses and row vs. column for the pairwise meta-analyses. Results are expressed as SMDs with 95% confidence intervals. A negative SMD indicates superiority of the first treatment over the comparison treatment. Bold estimates indicate a significant statistical difference. N = number of studies in the comparison. NA^1^ = no evidence on I^2^ was available as there was only one study for that comparison. NA^2^ = no study compared exercise vs. temazepam.

## Data Availability

Data collected in this study are available upon request.

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
