# Peer review of "Effectiveness of Exercise, Cognitive Behavioral Therapy, and Pharmacotherapy on Improving Sleep in Adults with Chronic Insomnia: A Systematic Review and Network Meta-Analysis of Randomized Controlled Trials"

_healthcare, 2023, doi:10.3390/healthcare11152207_

Round 1
Reviewer 1 Report
Thank you for the opportunity to review submission number 2477476 entitled “Long-term effectiveness of exercise, cognitive behavioral therapy and pharmacotherapy on improving sleep in adults with chronic insomnia: a systematic review and network meta-analysis of randomized controlled trials”. The present article examines the effectiveness of CBT, exercise and medication for managing chronic insomnia by pooling and analyzing results from RCTs with follow up time points at or beyond 6 months following the initiation of the intervention. The article is well written, and the authors support the need for this study well. However, as the follow-up durations in the pooled RCTs ranged from 6 months to 24 months, I believe the phrase “long term” is overstating these findings a bit. Modifying the language in the title to more accurately reflect and describe the follow up time period the results are able to speak to would be recommended.
Other comments
· Was there any consideration given to including studies with objective measures as well?
· The authors mention that the interventions ranges from 5-16 weeks—it would be helpful to know if possible if the participants in the RCTs continued practicing the skills they learned (or medications they were prescribed) independently after the study had ended but before the follow up time point. I think this is especially important for CBT-I and exercise, as the ongoing use of learned skills and practice would likely impact findings.
· The authors mention that major a limitation of the exercise group was that 2 of the 3 studies examining exercise examined tai chi. Did the authors consider changing the study inclusion criteria to create a greater balance between the nodes in the network? (e.g., more med studies, more exercise studies included).
· Table 2 – I find this formatting difficult to follow; please consider altering the layout
Reviewer 2 Report
Manuscript entitled: Long-term Effectiveness of Exercise, Cognitive Behavioral Therapy, and Pharmacotherapy on Improving Sleep in Adults with Chronic Insomnia: A Systematic Review and Network Meta analysis of Randomized Controlled Trials
I personally appreciated the authors hard work in preparing this manuscript for consideration. The manuscript is indeed well organized. However, there are some minor questionable issues raised in my mind, as follows:
- The reviewed studies were too small (only 13 studies, but 1350 participants were included in the study). It might not be clinically finalized if the author concluded that the use of both exercise and CBT-I for long-term management of chronic insomnia, while temazepam may be used for short-term treatment, However, it would be better if the authors provided more information to support the small amount of sample analysis.
- The authors mentioned that they systematically searched MEDLINE, PsycINFO, Embase, and SPORTDiscus for relevant articles published from database inception to January 30, 2022. I have some issues that might need the authors clarification. Firstly, what does the database's inception here mean? In what year were published articles counted? Secondly, why do they search for those relevant articles in MEDLINE, PsycINFO, Embase, and SPORTDiscus? Do they consider those relevant articles in Web of Science (ISI) and Scopus with Q1 and Q2 ranking journal databases? If they include these two databases, there might be more samples to be analyzed.
- The authors omitted to mention whether the ethical committee had approved all the selected studies. Since it was related to patients, only those studies with ethical committee approval were included in the analysis and systematic review.
If the authors are able to add more information concerning the above minor questionable issues, this manuscript will be appropriated for publication.
Reviewer 3 Report
1. The manuscript needs more literature review on the general benefits from long-term exercise, CBT, and Pharmacotherapy. Those are the dependent variables for the study.
2. In Introduction, more strong problem statement and justification for the study should be expanded.
3. More explanations should be expanded in data search method. For example, who are the two independent researchers? who is the third researcher?
4. Conclusion should be expanded.
Round 2
Reviewer 3 Report
The manuscript has revised many parts based on the reviewer's comment. However, still there should be needed to expand systematic literature review on the benefits from exercise in the Introduction.
